# Non-Diabetic Kidney Disease in Type 2 Diabetes Mellitus: A Changing Spectrum with Therapeutic Ascendancy

**DOI:** 10.3390/jcm12041705

**Published:** 2023-02-20

**Authors:** Narayan Prasad, Vamsidhar Veeranki, Dharmendra Bhadauria, Ravi Kushwaha, Jeyakumar Meyyappan, Anupama Kaul, Manas Patel, Manas Behera, Monika Yachha, Vinita Agrawal, Manoj Jain

**Affiliations:** 1Department of Nephrology, Sanjay Gandhi Postgraduate Institute of Medical Sciences, Lucknow 226014, India; 2Department of Pathology, Sanjay Gandhi Postgraduate Institute of Medical Sciences, Lucknow 226014, India

**Keywords:** non-diabetic kidney disease, diabetic kidney diseases, histological changes, type 2 diabetes mellitus

## Abstract

Background and objectives: Owing to changing epidemiology and therapeutic practices, a change in the spectrum of renal involvement in Type-2 diabetes mellitus (T2DM) has also been noted. The treatment of non-diabetic kidney disease (NDKD) differs from diabetic kidney disease (DKD) and the reversibility of NDKD in many cases to normal, prompts biopsy for rapid and accurate diagnosis. Data are scarce on kidney biopsy findings in T2DM. Study design & setting: In this observational study, we prospectively collected the data of kidney biopsies of patients aged ≥ 18 years with T2DM admitted between 1 August 2005 and 31 July 2022. The clinical, demographic and histopathological data were evaluated. The spectrum of kidney involvement in the form of DKD and/or NDKD was studied. The impact of these findings with the use of drugs retarding disease progression was also analyzed. Results: A total of 5485 biopsies were performed during the study period and of these 538 patients had T2DM. The mean age of the study population was 56.9 ± 11.5 years and 81% were males. The mean duration of DM was 6.4 ± 6.1 years. Diabetic retinopathy (DR) was noted in 29.7%. The most common indication for biopsy was an acute rise in creatinine (147, 27.3%). Amongst the 538 diabetic patients who underwent biopsy, histological features only of DKD were noted in 166 patients (33%), NDKD alone in 262 (49%) and NDKD with DKD lesions in 110 (20%). On multivariate analysis, duration of DM less than 5 years, absence of CAD, absence of DR, oliguria at presentation, an acute rise in creatinine and low C3 were associated with NDKD. Conclusions: The prevalence of NDKD among diabetics and ATIN in particular might be on an increasing trend in the current era of changing T2DM epidemiological patterns. The use of anti-pro-teinuric agents was associated with lesser degrees of histopathological chronicity in T2DM.

## 1. Introduction

Diabetes Mellitus (DM) affects the kidney either in the form of diabetic kidney disease (DKD), non-diabetic kidney disease (NDKD), or both overlapping [1]. Despite an increasing prevalence of type-2 diabetes mellitus (T2DM), the overall prevalence of DKD remained stable [2]. Apart from better glycemic control by early diagnosis, this could be due to the improved understanding of the pathogenesis, at least of the hemodynamic mechanisms, which led to the discovery and use of newer therapeutic agents, including renin-angiotensin-aldosterone system blockers (RAASB) and sodium-glucose transporter-2 inhibitors (SGLT2I). Recently, an increasing prevalence of NDKD in diabetics has been reported [3], which may be due to the aging population, higher exposure to infections, increasing incidence of monoclonal gammopathies and other malignancies [3,4]. Consequently, an increased prevalence of NDKD and non-proteinuric DKD is expected to rise in T2DM patients. More importantly, many NDKD conditions are treatable and complete reversibility is possible, which warrants a timely biopsy, accurate diagnosis and prompt treatment.

In the face of these epidemiological changes, histological and clinical characteristics are also likely to change; hence a re-evaluation of kidney disease in T2DM is implicated. Sharma SG et al. found an increasing prevalence of NDKD and a changing spectrum of NDKD [4]. For instance, acute tubular necrosis (ATN) was the most common type of NDKD found in contrast to IgA Nephropathy, as noted in an earlier meta-analysis [5]. Moreover, due to relatively poor glycemic and blood pressure control, delayed diagnosis and higher infection rates in the developing world, the NDKD spectrum differs from western studies and cannot be extrapolated to the Indian population [6]. Given the huge burden of diabetes in India, which accounts for nearly one-fifth of the diabetic population in the world, the burden of diabetes-related kidney diseases is also expected to be high [7]. Hence, knowing the spectrum of kidney diseases in T2DM is essential for optimal management. The current study attempts to analyze clinicopathological characteristics of various kidney diseases along with differences in findings with RAASB and SGLT2I treatment in patients with T2DM admitted for a kidney biopsy.

## 2. Materials and Methods

In this study, we prospectively collected data of all kidney biopsies performed in the department between the study period from 1 August 2005, till 31 July 2022. Written and informed consent was obtained from each patient for the kidney biopsy and inclusion into the study at time of the kidney biopsy. The study was conducted as per the declaration of Helsinki. The institutional ethics committee has approved the study.

We separated the biopsy details and clinical features at the presentation of the patients on the electronic records of the hospital information system of this public sector tertiary care institute. All patients had standard indications for biopsies, i.e., absence of retinopathy, presence of active sediments, red blood cells/cast in urine, heavy proteinuria on first presentation and sudden deterioration in degree of proteinuria, rapid decline in GFR as per standard guidelines [8,9]. Patients with Type-1 DM and pregnant and lactating females were excluded.

The indications of biopsies were prospectively noted. The data included demographic details (age, gender, duration of diabetes), associated comorbidities, diabetes-related microvascular and macrovascular complications, type of renal presentation (Acute kidney injury/Acute on CKD/Rapidly progressive renal failure (RPRF)), eGFR at presentation, dialysis dependency, proteinuria, presence of microscopic hematuria, hypoalbuminemia, hypercholesterolemia, hypocomplementemia and the physician’s indication of biopsy. Therapeutic parameters were collected, including therapy with RAASB, SGLT2I, insulin, other anti-hypertensives and glycemic control before the biopsy. The details of standard definitions, disease classification and the disease stages are described in the Appendix A. The definitions of DM, acute kidney injury (AKI), acute on chronic kidney disease (AKI on CKD), rapidly progressive renal failure (RPRF) and rapid decline in GFR were considered as per the standard diagnostic criteria shown in previous studies and briefed in the Appendix A [10,11,12,13,14,15,16]. Similarly, comorbidities such as hypertension, coronary artery disease, cerebrovascular and peripheral vascular disease, diabetic retinopathy and diabetic neuropathy were defined as per the standard disease definitions [17,18,19,20,21,22].

All biopsy samples were processed for light microscopy, immunofluorescence and electron microscopy if indicated [19]. Two pathologists, VA and MJ, reviewed histological changes in DKD, NDKD and DKD plus NDKD, as described in the Appendix A. Disputed findings were resolved by mutual agreement.

The DKD and various NDKD were classified as per the standard diagnostic criteria shown in the Appendix A [23,24,25,26,27,28,29,30]. Patients were categorized into three groups—DKD, NDKD and NDKD plus DKD groups. Briefly, the presence of features such as Kimmelstiel–Wilson (KW) lesions, mesangial expansion, capsular drops, fibrin caps and the absence of any other features of NDKD was defined as DKD. Pure NDKD was defined by the presence of predominantly vasculopathy, interstitial fibrosis, tubular atrophy and/or specific glomerular changes in the absence of classical changes of DKD. The histological pictures of the presence of both were defined as DKD plus NDKD. The clinical predictors of NDKD were determined. The clinical and histological findings were compared between DKD, NDKD and DKD plus NDKD. We also compared the histopathological features of patients on RAASB and/or SGLT2I versus those not on the same drugs.

### Statistical Analysis

Statistical analysis was performed by IBM, SPSS software, version 25. The Shapiro-Wilk test was used to determine the normality of the distribution of data values. All the continuous data were expressed in the form of mean and standard deviation if data were normally distributed, and the non-normally distributed data were expressed in the median and interquartile range. Categorical data were expressed in percentages. The Chi-square test or Fischer’s exact test was used to compare the categorical values between the groups, as per the application required. Student’s *t*-test was used to compare the mean values and continuous variables if it was normally distributed; otherwise, Mann-Whitney’s U-test was used. A stepwise multivariate logistic regression analysis was used to analyze the clinicopathological factors predicting NDKD. A *p*-value of <0.05 was considered significant.

## 3. Results

A total of 5485 biopsies were performed during the study period; of these, 538 (mean age 56.9 ± 11.5 years and 436 (81%), males) patients had T2DM (Figure 1). The baseline characteristics of the study population are as shown in Table 1. The mean duration of T2DM was 6.4 ± 6.1 years. Coexisting hypertension (HTN) was seen in 419 (78%) patients and of these only 244 (45.4%) patients had HTN onset after the onset of T2DM. Among the diabetes-related other end-organ changes, DR was most common, seen in 160 patients (29.7%); of these, most (122 patients, 76.2%) had non-proliferative DR. A total of 39.8% had nephrotic syndrome at presentation, 34% patients had AKI (including acute on CKD) and 7.8% had RPRF at presentation. The median eGFR at presentation was 17.2 (IQR: 9.5–41.7) mL/min/1.73 m^2^ and 29.2% had dialysis requiring renal failure. The mean proteinuria was 4.7 ± 3.9 g/day, with microscopic hematuria in 46.3% of patients.

### 3.1. Indication and Histological Findings of Biopsies

The indications of kidney biopsies are shown in Figure 2. The most common indication for biopsy was AKI (with or without an underlying CKD) in 151 (28%) patients, followed by the first presentation of nephrotic syndrome or rapidly rising proteinuria in 123 (22.8%) patients. Among the 538 diabetic patients who underwent biopsy (Figure 3), histological features of only DKD were noted in 166 patients (31%), pure NDKD in 262 patients (49%) and NDKD plus DKD lesions in 110 (20%) patients. Of the 372 patients with NDKD pathology, the most common NDKD pathology was acute tubulointerstitial nephritis (ATIN) in 126 (33.8%) patients (Figure 4A,B), followed by infection-related glomerulonephritis(IRGN) in 52 (13.9%) patients (Table 2). Membranous Nephropathy, IgA Nephropathy, Focal segmental glomerulosclerosis (FSGS) and membranoproliferative glomerulonephritis (MPGN) were reported in 36 (9.6%), 34 (9.1%), 29 (7.8%) and 17 (4.5%) patients, respectively. Crescentic glomerulonephritis with immune complex deposits was observed in 4% and pauci-immune crescentic glomerulonephritis in 3.7% of cases. Chronic tubulointerstitial nephritis was observed in 1.6%. Other lesions such as monoclonal immune deposition disease, thrombotic microangiopathy, amyloidosis, granulomatous interstitial nephritis and anti-glomerular basement membrane disease were seen in <5% of cases.

### 3.2. Clinicopathological Comparison between DKD and NDKD

A comparison of the clinicopathological features between patients with DKD alone and those with NDKD lesions on histopathology is shown in Table 3. A significantly higher number of patients with NDKD had a duration of DM <5-years (66.3% vs. 44%, *p* = 0.001). The mean duration of T2DM in DKD patients was significantly higher compared to those with NDKD (8.4 ± 6.6 years vs. 5.5 ± 5.4 years, *p* < 0.001). A significantly higher number of DKD patients had hypertension (86.1% vs. 74.2%, *p* = 0.001), CAD (11.4% vs. 5.6%, *p* = 0.01) and DR (45.8% vs. 22.5%, *p* = 0.001) compared to those with NDKD. Oliguria (20.4% vs. 46.5%, *p* = 0.001), evidence of recent infection within the past 4 weeks (19.6% vs. 7.8%, *p* = 0.001) and presence of extra-renal manifestations (19% vs. 9.6%, *p* = 0.008) were significantly higher among those who had NDKD lesions. A significantly higher number of patients had AKI (39% vs. 2.4%, *p* = 0.001) and RPRF (9.1% vs. 2.4%, *p* = 0.005) presentations in the NDKD group. The eGFR at presentation was similar between the two groups (18.9 ± 28.3 vs. 16.4 ± 23.1, *p* = 0.08) and 34.1% in the NDKD group required dialysis compared to 18% in DKD (*p* < 0.001). The mean proteinuria at presentation was significantly higher in the DKD group (5.9 ± 4.4 vs. 4.1 ± 3.5, *p* = 0.001). However, nephrotic syndrome was observed in a similar proportion of patients in both groups (41.5% vs. 44.8%, *p* = 0.83). A significantly higher proportion of patients with NDKD had microscopic hematuria (50.2% vs. 37.3%, *p* = 0.01). Similarly, a significantly higher number of patients with NDKD had low C3 (19.8% vs. 4.8%, *p* = 0.001) and low C4 (4.5% vs. 2.4%, *p* = 0.04). On multivariate logistic regression analysis (Table 4), the duration of DM < 5-years, absence of CAD, absence of DR, oliguria at presentation, an acute rise in creatinine and low C3 at presentation, were associated with NDKD.

### 3.3. Histological Comparison between DKD and NDKD

The histopathological features are shown in Appendix A. Comparing the vascular and tubulointerstitial changes on renal biopsy between the two groups (Appendix A), a significantly higher proportion of NDKD patients had no IFTA (15.8% vs. 1.8%, *p* = 0.001) than DKD patients. A higher proportion of DKD patients had a moderate degree of IFTA (47% vs. 29.8%, *p* = 0.001), suggesting a higher degree of chronicity among the patients with DKD (Figure 4C). Among the vascular changes, both afferent and efferent arteriolar hyalinosis was noted in 84.3% of DKD patients compared to 23.9% of NDKD patients (*p* = 0.001). The majority of patients in the NDKD group had either no hyalinosis (38.9% vs. 7.2%, *p* = 0.001) or only afferent arteriolar hyalinosis (37% vs. 8.4%, *p* = 0.001) as compared to DKD patients. On the evaluation of the immunofluorescence examination findings, 46.4% of patients with DKD had no immune deposits compared to 11.5% among the NDKD group (*p* = 0.001). The majority of NDKD patients had C3 deposits (31.4% vs. 15%, *p* = 0.001).

### 3.4. Histological Comparison between DKD and NDKD plus DKD

The comparison of histological classes of DKD and the diabetes-related glomerular changes in the DKD-only group and those with NDKD + DKD are shown in Appendix A. There were no differences in the distribution of various classes of DKD. Eight patients (4.8%) with DKD and two patients (1.8%) with NDKD plus DKD had no diabetes-related glomerular changes on light microscopy (*p* = 0.19). They had diabetes-related GBM thickening on EM only (Figure 4D). Nearly half of the patients (49%) among the NDKD plus DKD group had GBM thickening compared to 29.5% among the DKD patients (*p* = 0.003). Likewise, a higher percentage of patients with NDKD plus DKD had severe mesangial expansion (22.7% vs. 9.3%, *p* = 0.001) and K-W lesions (47.2% vs. 33.1, *p* = 0.03), compared to DKD only group. Most of the patients with only DKD had mild mesangial expansion (70.4% vs. 40.9%, *p* = 0.001).

### 3.5. Association of RAASB and SGLT2I Treatment with Histological Lesions

The comparison of the therapeutic effects of RAASB and combination therapy (RAASB plus SGLT2I) among DKD and NDKD is shown in Table 5. A total of 44 patients (26.5%) in DKD and 83 (22.3%) in NDKD were on RAASB (*p* = 0.28); 38 (22.8%) in DKD and 46 (12.3%) patients in NDKD groups were on combination therapy (*p* = 0.001). The median duration of RAASB therapy before the biopsy was 14.3 months (IQR: 3.4–66.8) in the DKD and 15.9 months (IQR: 4.7–26.3) in NDKD group (*p* = 0.3). The median duration of combination therapy was 8.1 months (IQR: 2.7–19.3) in the DKD group and 7.9 months (IQR: 3.5–19.2) in NDKD (*p* = 0.43).

Most patients with DKD on RAASB (72.7% vs. 32.8%, *p* <0.001) had mild IFTA and without RAASB had moderate to severe IFTA (66.4% vs. 25%, *p* < 0.0001). Likewise, a higher proportion of DKD patients had mild IFTA (73.6% vs. 53.9%, *p* = 0.03) with combination therapy and moderate to severe IFTA (45.3% vs. 23.6%, *p* =0.01) without combination therapy.

NDKD patients on RAASB (33.7% vs. 10.7%, *p* < 0.001) and combination therapy (45.6% vs. 11.6%, *p* < 0.001) had no IFTA. A higher proportion of the NDKD cohort without RAASB (40% vs. 21.7%, *p* = 0.002) or without combination therapy (43.8% vs. 4.3%, *p* < 0.001) had moderate to severe IFTA. NDKD patients on combination therapy had fewer vascular hyalinotic changes (n = 17, 36.9%) than those without combination therapy (n = 210, 64.4%; *p* = 0.001). However, no significant therapeutic effects were observed with respect to the vascular changes in DKD.

## 4. Discussion

In this study, we analyzed clinical, laboratory and pathological features in 538 T2DM patients who underwent kidney biopsies for various indications. This accounted for one-tenth of all the biopsies performed at our center. While nearly one-third had only DKD, 70% had NDKD, of which 50% had pure NDKD and 20% had NDKD with DKD. Given the high pre-test probability of NDKD, a higher percentage of NDKD was expected, reiterating the findings of Sharma et al. [4]. On the contrary, nearly a third of diabetic patients with atypical clinical features eventually had DKD on biopsy.

We found DR only in 45% of patients with DKD compared to more than 60% in previous studies [31,32], which suggests the declining correlation between the two micro-vasculopathies, similarly evidenced by Pedro et al., who found DR in <30% of T2DM with DKD [33]. More importantly, up to 80% of the patients without DR eventually had NDKD. Similarly, many recent studies also showed that 70–80% of diabetic patients with NDKD had no DR [34,35]. Accordingly, “the renal-retinal relation” in T2DM may be better asserted, as the absence of DR may reasonably point towards NDKD. However, the presence of DR does not predict DKD, as also evidenced in other previous studies [36,37]. Similarly to this one no correlation between diabetic neuropathy and DKD was found in a previous study [38].

Coexisting macrovascular complications were noted in more than 10% of DKD patients. We observed lesser numbers of CAD in our study compared to prior studies, where about 60% of diabetic patients with CKD had CAD [39,40]. This could be due to the cross-sectional nature of the data retrieved at the time of the kidney biopsy and the lack of follow-up. Despite the good evidence that CKD in diabetes was associated with an increased CAD risk, whether DKD is associated with higher CAD risk compared to NDKD remains debatable. We found that, among diabetics with kidney failure, the presence of DKD, in particular, doubles the risk of CAD compared to NDKD (12% vs. 6%). Microscopic hematuria was observed in 40% of DKD patients, similar to prior studies in 20–40% of patients [41]. Non-proteinuric kidney disease was noted more in the NDKD group; only about 5% of DKD patients had a non-proteinuric phenotype in our study. Despite the increasing prevalence of non-proteinuric DKD phenotype currently, [42,43], less prevalence in our study could probably be because this phenotype might not have been considered for kidney biopsy. We also observed that two-thirds of patients with NDKD had diabetes duration of <5 years and up to 40% with T2DM had a pre-existent HTN, as observed in other studies [44,45].

Moreover, a significantly higher proportion of DKD patients had an onset of HTN after at least two years of diagnosis of T2DM. Our study emerged with strong clinical evidence that diabetes duration <5 years, absence of CAD, absence of DR, oliguria at presentation, an acute rise in creatinine and low C3 were predictors of NDKD on multivariate analysis. Microscopic hematuria, HbA1C level, age, gender, etc., were the other factors noted to be higher in NDKD patients, although non-significant on multivariate analysis.

Amongst NDKD, ATIN was the most common pathology in our study, in concordance with evidence in the past decade [4,46,47], unlike in prior studies, where glomerular lesions, such as IgAN and MGN, were more common NDKD pathologies [48,49]. A meta-analysis by Fiorentino et al. further supported our findings of the increasing prevalence of ATIN in recent studies [5]. The changing pathology of NDKD could be partially attributed to epidemiological changes, the aging population and increased infections and malignancy risk. Besides, nephrotoxic medications and herbal remedies exposures may also contribute to ATIN in a pre-existent state of diabetes-related vascular ischemia. The types of NDKD pathology may also depend on study inclusion criteria and indications for biopsy, besides demographic and genetic variations in the study population. We performed only clinically indicated biopsies on patients with a high suspicion of NDKD; therefore, outcomes may differ from research-indicated biopsies. Hence, as the most common indication for biopsy in our study, AKI, either with (23%) or without CKD (7%), ATIN and proliferative GN contributed to more than 50% of the NDKD pathologies. However, membranous nephropathy and IgA nephropathy were the common pathologies amongst non-AKI indications groups, in concurrence with previous studies. Our finding for NDKD pathologies emphasized the need for a high index of clinical suspicion and for kidney biopsies in the current period when evaluating renal dysfunction in diabetics.

The study’s strength lies in the large sample size and prospective data collection during biopsy. Our study is plausibly the first to evaluate the clinicopathological findings predicting NDKD in the best possible way in a single study and conceivably the first to analyze the association between RAASB therapy and/or SGLT2I therapy with histopathological findings in both DKD and NDKD patients. Data on the impact of SGLT2I on histological findings are currently scarce. Major limitations are the lack of long-term follow-up data post-biopsy, and selection bias for indication biopsy remains unavoidable.

## 5. Conclusions

To conclude, the prevalence of NDKD among diabetics and ATIN, in particular, might be on an increasing trend in the current era of changing epidemiology and therapies for T2DM. Because of the treatable nature of the condition and its potential to accelerate progression with delayed diagnosis of renal dysfunction, a greater emphasis on timely diagnosis is required for this particularly vulnerable cohort. The spectrum of NDKD is vast and may be missed without a kidney biopsy.

## Figures and Tables

**Figure 1 jcm-12-01705-f001:**
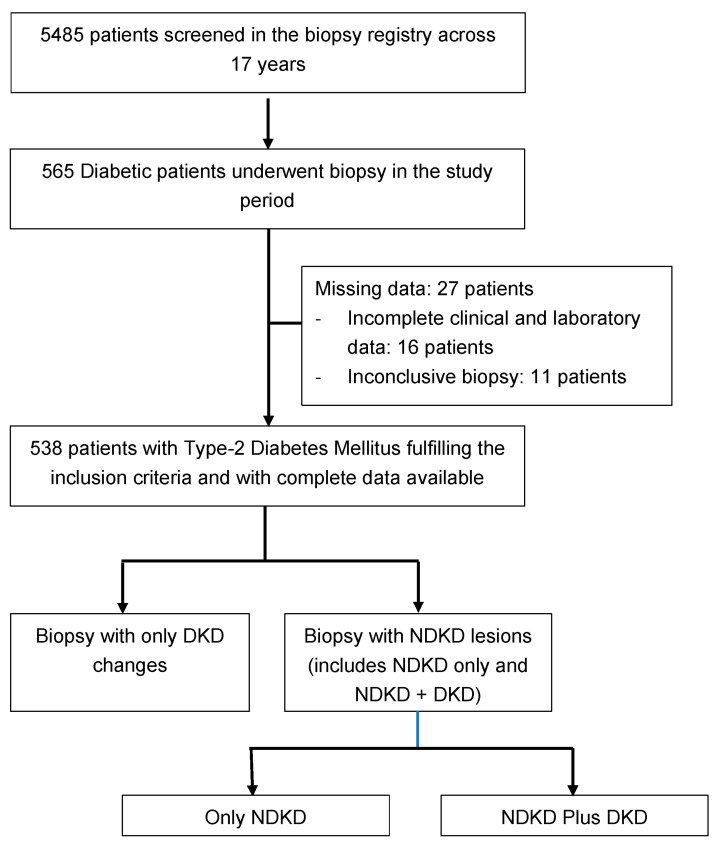
Study methodology. DKD: Diabetic Kidney disease; NDKD: Non-diabetic kidney disease.

**Figure 2 jcm-12-01705-f002:**
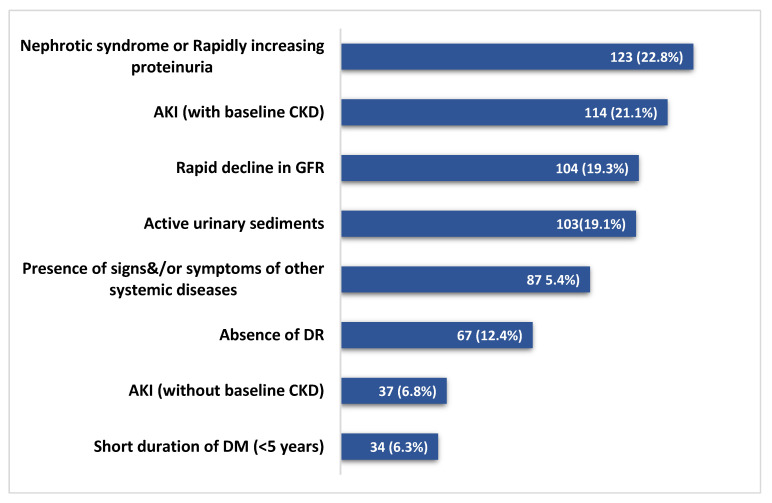
Indications for renal biopsy. AKI—Acute kidney injury; Rapid decline in GFR: Decline in GFR at a rate of >5 mL/min/1.73 m^2^ per year; GFR: Glomerular filtration rate; Active urinary sediments: >3 erythrocytes per high power field; DR: Diabetic retinopathy; CKD: Chronic kidney disease; DM: Diabetes mellitus. Note: Few patients have more than one indication for biopsy.

**Figure 3 jcm-12-01705-f003:**
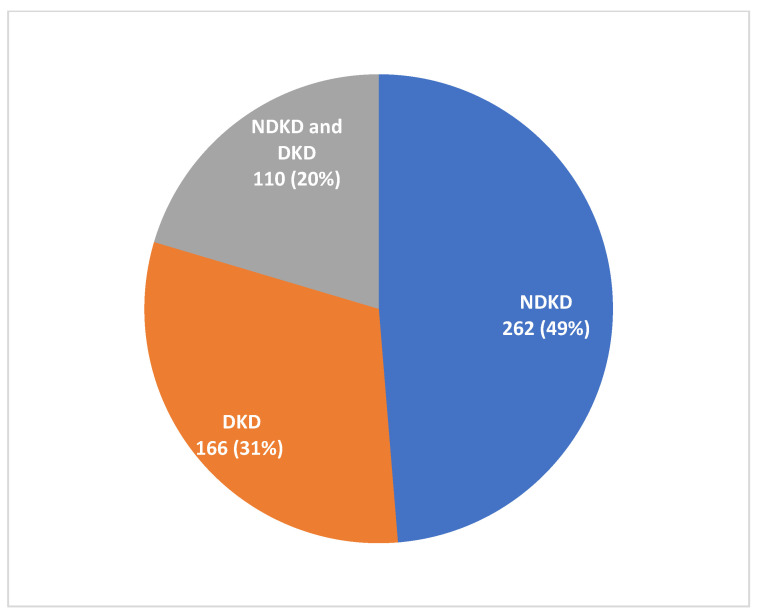
Frequencies of Various histological classes on Kidney biopsy. DKD: Diabetic Kidney disease; NDKD: Non-diabetic kidney disease.

**Figure 4 jcm-12-01705-f004:**
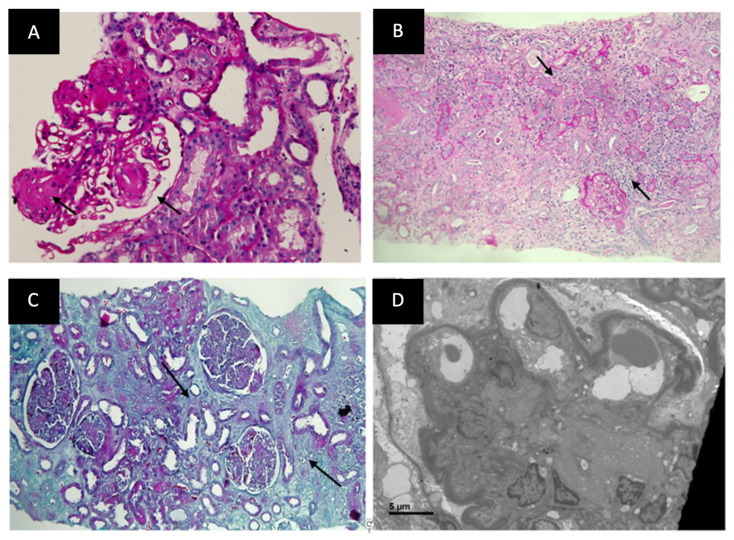
Histopathological findings of case examples with diabetic and non-diabetic renal changes. (**A**) Renal biopsy shows KW nodule (arrow) with aneurysmal dilatation of peripheral glomerular capillary loops. PAS stain ×400 original magnification; (**B**) Renal biopsy from a patient of diabetic nephropathy showing mixed tubulointerstitial inflammation (arrows); (**C**) Section showing moderate IFTA (arrows) in a patient of diabetic nephropathy. Masson Trichrome stain ×100 original magnification; (**D**) Electron microscopy image in a patient of diabetic nephropathy shows KW nodule, peripheral patent capillary loops, thickened GBM and podocytopathy.

**Table 1 jcm-12-01705-t001:** Baseline characteristics (N = 538).

Variable	Frequency
Age (years, mean ± SD)	56.9 ± 11.5
Gender - Male (%) - Female (%)	436 (81%) 102 (19%)
Diabetes Mellitus duration (years, mean ± SD)	6.4 ± 6.1
Glycemic control (HbA1C%) - Fair glycemic control (<8%) - Poor Glycemic control (>8%)	230 (42.7%) 308 (57.3%)
Other comorbidities - HTN (%) - Hypothyroidism (%) - BA/COPD (%)	419 (78%) 72 (13.3%) 23 (4.2%)
Uncontrolled hypertension (n, %)	158 (29.4%)
Hypertension duration (years, mean ± SD)	3.8 ± 4.9
Hypertension onset in relation to Diabetes - HTN after DM - HTN before DM	244 (45.4%) 197 (36.6%)
Macrovascular complications - CAD - CVA - PAD	40 (7.4%) 17 (3.2%) 16 (3%)
Microvascular complications - Diabetic retinopathy - Diabetic neuropathy	160 (29.7%) 26 (4.8%)
Presenting complaints - Edematous illness - Oliguria - Microscopic hematuria (>3 RBC/HPF) - Evidence of recent infection	496 (92.2%) 207 (38.5%) 16 (3%) 86 (16%)
Syndromic presentation - Nephrotic syndrome - Acute Kidney Injury - Rapidly progressive renal failure	214 (39.8%) 183 (34%) 42 (7.8%)
eGFR at presentation (median, IQR); mL/min/1.73 m^2^	17.2 (9.5–41.7)
Dialysis requiring renal failure at presentation (%)	157 (29.2%)
Proteinuria at presentation (g/day), mean (±SD)	4.7 ± 3.9
Microscopic hematuria (>3 RBC/HPF)	249 (46.3%)
Laboratory parameters - Hb (g%, Mean_SD) - TLC (/μL, Mean_SD) - Platelet count (/μL, Mean_SD) - Serum albumin (g/dL, Mean_SD)	9.8 ± 2.3 9596 ± 3809 1.3 ± 1.1 3.2 ± 0.7

BA, Bronchial Asthma; COPD, Chronic Obstructive Pulmonary Disease; HTN, Hypertension; DM, Diabetes Mellitus; TLC, Total leukocyte count; CAD, Coronary artery disease; CVA, Cerebrovascular Accident; PAD, Peripheral artery disease; Microscopic hematuria, >3 erythrocytes per high power field; Evidence of recent infection within past 4–6 weeks—history or examination evidence supported by laboratory evidence (e.g. Anti-streptolysin titer).

**Table 2 jcm-12-01705-t002:** Various histological types of non-diabetic kidney diseases (N = 372).

Renal Biopsy Diagnosis	Total (N = 372)	NDKD Only (n = 262)	NDKD and DKD (n = 110)
ATIN (a) AIN (b) ATN	126 (33.8%) 79 (21.2%) 47 (12.6%)	70 (26.7%) 42 (16%) 28 (10.7)	56 (50.9%) 37 (33.6%) 19 (17.2%)
IRGN	52 (13.9%)	32 (12.2%)	20 (18.2%)
Membranous nephropathy	36 (9.6%)	34 (12.9%)	2 (1.8%)
IgA nephropathy	34 (9.1%)	29 (11.1%)	5 (4.5%)
FSGS (a) Primary FSGS (b) Secondary FSGS	29 (7.8%) 15 (4%) 14 (3.7%)	27 (10.3%) 15 (5.7%) 12 (4.5%)	2 (1.8%) 0 2 (1.8%)
MPGN	17 (4.5%)	13 (4.9%)	4 (3.6%)
Cast Nephropathy	15 (4%)	13 (4.9%)	2 (1.8%)
Crescentic GN—Immune complex associated	15 (4%)	5 (1.9%)	10 (9.1%)
Pauci-immune Crescentic GN	14 (3.7%)	14 (5.3%)	0
MCD	8 (2.1%)	8 (3.1%)	0
Granulomatous interstitial nephritis	6 (1.6%)	4 (1.4%)	2 (1.8%)
Chronic interstitial nephritis	6 (1.6%)	2 (0.7%)	4 (3.6%)
Lupus nephritis	3 (0.8%)	3 (1.1%)	0
Amyloidosis	4 (1%)	4 (1.4%)	0
Thrombotic microangiopathy	3 (0.8%)	2 (0.7%)	1 (0.9%)
Monoclonal immune deposition disease	2 (0.5%)	1 (0.4%)	1 (0.9%)
Anti-GBM disease	2 (0.5%)	1 (0.4%)	1 (0.9%)

ATIN: Acute tubule-interstitial nephritis; ATN: Acute tubular necrosis; AIN: Acute interstitial Nephritis; IRGN: Infection-related glomerulonephritis; FSGS: Focal Segmental Glomerulosclerosis; MPGN: Membranoproliferative glomerulonephritis; Pauci-immune crescentic GN: Crescentic glomerulonephritis—Both ANCA positive and ANCA negative; MCD: Minimal change disease; Anti-GBM disease: Anti-glomerular basement membrane disease.

**Table 3 jcm-12-01705-t003:** Comparison of patients with diabetic kidney disease and non-diabetic kidney disease (N = 538).

Variable	DKD Only (n = 166)	NDKD (n = 372)			*p*-Value
		†NDKD Only (n = 262)	††NDKD + DKD (n = 110)	Total (n = 372) († + ††)	
Age (years, mean ± SD)	55.4 ± 11.2	57.2 ± 12.0	58.5 ± 10.3	56.9 ± 11.5	0.06
Gender - Male (%) - Female (%)	131 (79%) 35 (21%)	212 (81%) 50 (19%)	88 (80%) 22 (20%)	305 (82%) 67 (18%)	0.79
Duration of Diabetes mellitus * (years, mean ±SD)	8.4 ± 6.6	5.5 ± 5.4	7.9 ± 6.5	5.5 ± 5.4	<0.001
Diabetes onset within 5 years	73 (44%)	193 (73.6%)	53 (48.2%)	247 (66.3%)	<0.001
Glycemic control (HbA1C%, Mean ± SD) Poor glycemic control (>8%)	6.8 ± 2.3 98 (59%)	6.8 ± 2.6 134 (51.1%)	6.4 ± 2.7 76 (69.1%)	6.7 ± 2.6 210 (56.4%)	0.96 0.22
Hypertension (%) Uncontrolled hypertension (%)	143 (86.1%) 49 (29.5%)	178 (74.1%) 57 (29.3%)	98 (89%) 47 (42.7%)	276 (74.1%) 109 (29.3%)	<0.001 0.81
Duration of hypertension (years, Mean ± SD)	4.1 ± 3.5	3.6 ± 2.8	3.9 ± 2.6	3.6 ± 4.8	0.35
HTN onset in relation to diabetes - HTN after DM - HTN prior to DM	92 (55.4%) 51 (30.7%)	87 (33.2%) 93 (35.4%)	50 (45.4%) 46 (41.8%)	137 (36.8%) 139 (37.3%)	<0.001 0.13
Diabetes related macrovascular complications - CAD - CVA - PAD	19 (11.4%) 4 (2.4%) 5 (3%)	15 (5.7%) 5 (1.9%) 3 (1.1%)	6 (5.4%) 8 (7.2%) 8 (7.2%)	21 (5.6%) 13 (3.5%) 11 (2.9%)	0.01 0.53 0.93
Diabetes related microvascular complications - Diabetic retinopathy - Diabetic Neuropathy	76 (45.8%) 12 (7.2%)	56 (21.3%) 9 (3.4%)	28 (25.4%) 5 (4.5%)	84 (22.5%) 14 (3.7%)	<0.001 0.07
Presenting complaints - Edematous illness - Oliguria - Macroscopic hematuria - Evidence of recent infection - Presence of extra-renal manifestations	155 (93.3%) 34 (20.4%) 4 (2.4%) 13 (7.8%) 16 (9.6%)	236 (90%) 111 (42.3%) 4 (1.5%) 54 (20.6%) 23 (8.7%)	105 (96.3%) 56 (50.9%) 8 (7.2%) 19 (17.2%) 18 (16.3%)	341 (91.6%) 173 (46.5%) 12 (3.2%) 73 (19.6%) 71 (19%)	0.09 <0.001 0.64 0.001 0.008
Syndromic presentation - Nephrotic syndrome - Acute Kidney Injury - Rapidly progressive renal failure	69 (41.5%) 4 (8.4%) 4 (2.4%)	84 (32%) 83 (31.6%) 23 (8.7%)	25 (22.7%) 62 (56.3%) 11 (10%)	167 (44.8%) 188 (50.5%) 34 (9.1%)	0.83 <0.001 0.005
eGFR at presentation (median, IQR); mL/min/1.73 m^2^	23.1 (11.5–45.5)	19.4 (10–51.5)	18.5 (9.1–22.3)	15 ± 23.1	0.08
Dialysis requiring renal failure at presentation	30 (18%)	75 (28.6%)	48 (43.6%)	127 (34.1%)	<0.001
Mean proteinuria at presentation (24-h urine protein in g/day, mean ± SD)	5.9 ± 4.4	4.2 ± 3.5	3.8 ± 3.4	4.1 ± 3.5	<0.001
Degree of proteinuria - <500 mg/day - 500 mg–3.5 g/day - >3.5 g/day	8 (4.8%) 60 (36.1%) 98 (59%)	27 (10.3%) 131 (51.5%) 123 (46.9%)	13 (11.8%) 61 (55.4%) 43 (39%)	170 (45.6%)	0.02 0.001 0.001
Microscopic hematuria (>3 RBC/HPF)	62 (37.3%)	136 (51.9%)	48 (43.6%)	187 (50.2%)	0.01
Laboratory parameters - Hb (g%, mean, SD) - TLC - Platelet count Serum albumin Lipid profile - Triglycerides - Total cholesterol Serum complement levels - Low C3 - Low C4	9.6 ± 2.2 9017 ± 3240 1.32 ± 1.06 3.22 ± 0.7 176 ± 89.6 185 ± 73.2 8 (4.8%) 2 (2.4%)	9.8 ± 2.4 9852 ± 4014 1.25 ± 1.13 3.19 ± 0.7 195 ± 118 189 ± 90.3 41 (15.6%) 16 (6.1%)	9.5 ± 1.8 9761 ± 4021 1.32 ± 0.96 3.2 ± 0.7 188.4 ± 98.5 176.4 ± 104.6 31 (28.2%) 1 (0.9%)	9.8 ± 2.4 9852 ± 4014 1.25 ± 1.13 3.19 ± 0.7 195 ± 118 189 ± 90.3 74 (19.8%) 17 (4.5%)	0.24 0.07 0.52 0.64 0.07 0.62 <0.001 0.04

* Duration of diabetes mellitus: significantly higher among the NDKD plus DKD group in comparison with NDKD only group (*p* < 0.001). Comparisons were performed between DKD and NDKD groups. NDKD consisted of pure NDKD, single dagger sign, plus NDKD with DKD showing double dagger sign in the table. The chi-square test or Fischer’s exact test was used to compare the categorical values between the groups, as per the application required. Student’s *t*-test was used to compare the mean values between two groups and continuous variables if it was normally distributed; else, Mann-Whitney’s U-test was used. Abbreviations: BA, Bronchial Asthma; COPD, Chronic Obstructive Pulmonary Disease; HTN, Hypertension; DM, Diabetes Mellitus; CAD, Coronary artery disease; CVA, Cerebrovascular Accident; PAD, Peripheral artery disease; Hb, Hemoglobin; TLC: Total Leukocyte count; Evidence of recent infection within the past 4–6 weeks—history or examination evidence supported by laboratory evidence (e.g., anti-streptolysin titer). NDKD include both pure NDKD, denoted by a single dagger sign; and NDKD plus DKD, denoted by a double dagger sign, the *p* value is derived for comparison between DKD versus NDKD, which is pure NDKD plus NDKD overlapping with DKD.

**Table 4 jcm-12-01705-t004:** Multivariate logistic regression analysis showing predictor of NDKD.

Variable	*p*-Value	Odds Ratio (95% Confidence Interval)
** Duration of DM ≤ ** **5 years**	**0.003**	**1.9 (1.26–3.14)**
Hypertension	0.22	0.73 (0.3–2.11)
Hypertension onset after diabetes	0.94	0.11 (0.03–1.86)
**Absence of CAD**	**0.05**	**2.1 (1.11–4.96)**
**Absence of DR**	**<0.001**	**4.9 (2.9–8.4)**
**Oliguria**	**0.02**	**1.8 (1.1–3.1)**
Evidence of infection (ongoing/recent past)	0.63	1.2 (0.52–1.44)
Presence of extra-renal manifestations	0.64	1.8 (0.82–2.26)
**Acute kidney Injury/Acute on Chronic kidney disease**	**<0.001**	**6.2 (3.4–11.0)**
Rapidly progressive renal failure	0.58	1.23 (0.6–2.68)
Dialysis requiring renal failure at presentation	0.06	1.79 (0.96–3.3)
Microscopic hematuria	0.07	1.51 (0.95–2.41)
Nephrotic range proteinuria	0.66	0.48 (0.2–1.8)
Sub-nephrotic proteinuria	0.52	1.18 (0.4–3.1)
**Low serum C3 level**	**<0.001**	**4.9 (2.08–11.6)**
Low serum C4 level	0.77	1.6 (0.8–4.6)

CAD: Coronary artery disease; DR: Diabetic retinopathy; Evidence of active infection/infection within recent past of 4–6 weeks—history or examination evidence supported by laboratory evidence (e.g., Anti-streptolysin titer). Acute rise in creatinine (AKI)—Defined as per the KDIGO 2012 guidelines of AKI i.e., Increase in serum creatinine by ≥0.3 mg/dL within 48 h (or) increase in serum creatinine to ≥1.5 times baseline, which is known or presumed to have occurred within the prior 7 days.

**Table 5 jcm-12-01705-t005:** Biopsy findings based on therapeutic differences.

**Diabetic Kidney Disease** **(n = 166)**	**RAASB**			**RAASB + SGLT2I**		
	**Yes** **(n = 44)**	**No** **(n = 122)**	***p*-Value**	**Yes** **(n = 38)**	**No** **(n = 128)**	***p*-Value**
Degree of IFTA - No IFTA - Mild - Moderate to severe IFTA	1 (2.3%) 32 (72.7%) 11 (25%)	1 (0.8%) 40 (32.8%) 81 (66.4%)	0.43 <0.001 <0.001	1 (2.6%) 28 (73.6%) 9 (23.6%)	1 (0.7%) 69 (53.9%) 58 (45.3%)	0.33 0.03 0.01
Vascular changes - No hyalinosis - Both afferent and efferent arteriolar Hyalinosis - Only afferent arteriolar hyalinosis	5 (11.3%) 34 (77.2%) 5 (11.3%)	7 (5.7%) 103 (84.4%) 12 (9.7%)	0.21 0.28 0.76	5 (13.1%) 18 (47.3%) 15 (39.5%)	7 (5.4%) 86 (67.1%) 35 (27.3%)	0.10 0.02 0.15
Non-diabetic kidney disease (n = 372)	RAASB			RAASB + SGLT2I		
	Yes (n = 83)	No (n = 289)	*p*-value	Yes (n = 46)	No (n = 326)	*p*-value
Degree of IFTA - No IFTA - Mild - Moderate to severe IFTA	28 (33.7%) 37 (44.5%) 18 (21.7%)	31 (10.7%) 131 (45.3%) 127 (40%)	<0.001 0.89 0.002	21 (45.6%) 23 (50%) 2 (4.3%)	38 (11.6%) 145 (44.4%) 143 (43.8%)	<0.001 0.47 <0.001
Vascular changes - No hyalinosis - Both afferent and efferent arteriolar hyalinosis - Only afferent arteriolar hyalinosis	38 (45.7%) 26 (31.3%) 19 (22.8%)	104 (35.9%) 63 (21.7%) 122 (43.9%)	0.10 0.07 <0.001	29 (63%) 4 (8.6%) 13 (28.2%)	116 (35.5%) 85 (26%) 125 (38.3%)	<0.001 0.009 0.18

Foot notes: A comparison was performed among DKD and NDKD groups between those on RAASB and not on RAASB. Similarly, those on combination RAASB+SGLT2i and not on the same, were also compared. Abbreviations: IFTA, Interstitial fibrosis and tubular atrophy; IFTA classification is as per the standard recommendations, including RPS classification, No IFTA: <10% Mild IFTA: 10–25%, Moderate IFTA: 25–50%; Severe IFTA: >50%. RAASB: Renin-angiotensin-aldosterone system blockers; SGLT2I: Sodium-glucose transporter 2 inhibitors.

## Data Availability

The data can not be public due to institution policy. However, it is available with the corresponding author and can be made available upon reasonable request.

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
