# Peer review of "Non-Diabetic Kidney Disease in Type 2 Diabetes Mellitus: A Changing Spectrum with Therapeutic Ascendancy"

_jcm, 2023, doi:10.3390/jcm12041705_

Round 1

Reviewer 1 Report

The manuscript entitled “Non-diabetic kidney disease in type 2 diabetes mellitus: A changing spectrum with therapeutic ascendancy” is an intrigue study about  kidney biopsy findings in T2DM.
Due to the lack of comprehensive study on this theme, publication is very innovative and is an indication for further research on this problem.

Although the manuscript requires a few corrections: the figures are very illegible, not all abbreviations in the tables and figures are explained. please verify and the numbers in figure 4 are not visible

Author Response

Comments by reviewer 1:

The manuscript entitled “Non-diabetic kidney disease in type 2 diabetes mellitus: A changing spectrum with therapeutic ascendancy” is an intrigue study about  kidney biopsy findings in T2DM.
Due to the lack of comprehensive study on this theme, the publication is very innovative and is an indication for further research on this problem.

Although the manuscript requires a few corrections: the figures are very illegible, not all abbreviations in the tables and figures are explained. please verify and the numbers in figure 4 are not visible.

Answer: Thanks to the reviewer for your appreciation. Explained all abbreviations and improved the figures.

Reviewer 2 Report

Dear Authors,

I have read your interesting article carefully and have the following remarks:

1) English needs some minor corrections

2) Page 2. Materials and Methods section, 3rd passage.

‘’…were considered as per the standard definitions considered in the study and described in the supplementary file (I).’’ There are numerous references cited in the Supplementary file (I), 10-30. The should be written also here, because they are not cited in the text elsewhere.

3) Section 2.1 Statistical analysis.

Here descriptions are very scarce. For non-normally distributed data the way of expression should be stated.

Also, for the comparison between groups, the methods you state (Student t test and Mann-Whitney U test, are for 2 groups. But, in some tables you compare 3 groups, and for those different tests are used.

Also, in the tables when you present the results of those comparison, the used test should be mentioned.

You write that you use multivariate cox regression analysis, but later when presenting results, you write logistic regression – these are different methods, not the same.

Also, you must describe in further detail the regression method used (stepwise, forward, backward?; which variables were included,…).

Which significance level was used?

4) Table 1. Hypertension duration does not have units. TLC is not explained.

5) Figure 2. The Figure title, description and legend should be written below the Figure. It says in this figure ‘’rapid decline in GFR’’, but in supplementary file (I) it says ‘’rapid fall in GFR’’ – the terminology must be unified throughout the text.

Also, regarding figure 2., ‘’rapidly increasing proteinuria’’ and ‘’active urinary sediments’’ are not defined.

6) Figure 3. Title should be written below the figure.

7) Table 3. Mean proteinuria at presentation does not have units nor clarification (24 hour?)

8) Supplementary files. Table 1. – Acute rise in creatinine is not defined. Table 3. Doesn’t have title

9) I think that the supplementary table 1 is very important and should be included in the main text.

Best regards

Author Response

Comments by reviewer 2:

  1. English needs some minor corrections

Answer: Corrections made as per suggestions

  1. Page 2. Materials and Methods section, 3rd passage.‘’…were considered as per the standard definitions considered in the study and described in the supplementary file (I).’’ There are numerous references cited in the Supplementary file (I), 10-30. The should be written also here, because they are not cited in the text elsewhere.

Answer: Corrections made, and the references are included in the text as suggested (Lines 85-93)

  1. Section 2.1 Statistical analysis. Here descriptions are very scarce. For non-normally distributed data the way of expression should be stated.

Answer: Statistical analysis was elaborated as per suggestions and highlighted in yellow. (Line 111-113 and 118-120)

  1. Also, for the comparison between groups, the methods you state (Student t test and Mann-Whitney U test, are for 2 groups. But, in some tables you compare 3 groups, and for those, different tests are used.

Answer:  The reviewer refers to table 3 to clarify the comparison between groups. In table 3, the comparison was made across two groups - between those with DKD (denoted with a single dagger sign) and those with NDKD ( denoted with a double dagger sign), which included pure NDKD plus NDKD with DKD lesions. It has also been clarified in the footnotes of the table. (line: 244, 245)

  1. Also, in the tables when you present the results of those comparison, the used test should be mentioned.

Answer: Details of statistical tests were included as per suggestions. (line 261-263)

  1. You write that you use multivariate cox regression analysis, but later when presenting results, you write logistic regression – these are different methods, not the same.

Answer: The statistical test used was multivariate logistic regression analysis and is corrected at appropriate sites. (Line 113-115)

  1. Also, you must describe in further detail the regression method used (stepwise, forward, backward?; which variables were included,…).

Answer: Stepwise regression method was used. (Line 113-115)

  1. Which significance level was used?

Answer: A p-value of < 0.05 was considered significant and included in the text (line 115).

  1. Table 1. Hypertension duration does not have units. TLC is not explained.

Answer: Added to the manuscript as per suggestions. (Table:3, Line 150 and line 266)

  1. Figure 2. The Figure title, description and legend should be written below the Figure. It says in this Figure ‘’rapid decline in GFR’’, but in the supplementary file (I) it says ‘’rapid fall in GFR’’ – the terminology must be unified throughout the text.

Answer: Decline in GFR at a rate of > 5ml/min/1.73m2 per year was defined as rapid decliner. The corrections were done as per suggestions (Line 75, Supplementary file: Section I(D).

  1. Also, regarding figure 2., ‘’rapidly increasing proteinuria’’ and ‘’active urinary sediments’’ are not defined.

Answer:  The definition of active urinary sediments (proteinuria with RBC >3 per hpf) in the urine is shown in line 177. Rapidly increasing proteinuria was considered if daily proteinuria was higher than expected, as per the physician.

  1. Figure 3. Title should be written below the Figure.

Answer: The title is added below the Figure (Line 183)

  1. Table 3. Mean proteinuria at presentation does not have units nor clarification (24 hour?)

Answer: It is 24-hour urine protein, and the same is mentioned in table 3

  1. Supplementary files. Table 1. – Acute rise in creatinine is not defined.

Answer: Acute rise in creatinine implies acute kidney injury, that is, 0.3mg/dl increase in creatinine within 48 hours and was described in the supplementary file. Further clarification was added to the footnote of table 4 (line 282-285)

  1. Table 3. Doesn’t have title

Answer: The title is added again.

  1. I think that the supplementary table 1 is very important and should be included in the main text.

Answer: supplementary Table 1 is included in the main manuscript (Table 4)

Reviewer 3 Report

In this article, the author, analyzed clinicopathological characteristics of various kidney diseases along with differences in findings with RAASB and SGLT2I treatment in patients with T2DM admitted for a kidney biopsy. The authors clearly describe the background highlighting how DKD and NDKD differ both in the treatment used and in the possible reversibility of the disease. For this reason, this scientific article studies the importance of early diagnosis using kidney biopsy. In addition to identifying DKD or DKD, this approach also allows to evaluate a particular disease status, i.e. NDKD + DKD and therefore to study a possible disease overlap. From a methodological point of view, the authors describe the type of study well, specifying that it is a prospective observational study and also specifying the timeline. Furthermore, the authors confirm the correctness of the procedures in terms of data protection, privacy and signing of the informed consent before enrolling the patients in the study. The use of statistical tests is also good, which are in line with the choice of data analysis but certainly with the data in possession and, evaluating the type of study, the authors could have used other statistical techniques to strengthen their initial hypothesis. the presentation of the tables and graphs is good but could be improved. The results show how important the prevalence of NDKD is and how important it is to be able to build an early diagnosis by biopsy in order to adapt the best possible pharmacological treatment and avoid a possible progression of the disease. The advice I give is to perform an analysis of the interaction versus gender variable, as the authors highlight in the descriptive table that 81% of the patients are male and gender could modify or interact with the pathophysiology of the patients studied.

Author Response

Comments by reviewer 3:

In this article, the author, analyzed clinicopathological characteristics of various kidney diseases along with differences in findings with RAASB and SGLT2I treatment in patients with T2DM admitted for a kidney biopsy. The authors clearly describe the background highlighting how DKD and NDKD differ both in the treatment used and in the possible reversibility of the disease. For this reason, this scientific article studies the importance of early diagnosis using kidney biopsy. In addition to identifying DKD or DKD, this approach also allows to evaluate a particular disease status, i.e. NDKD + DKD and therefore to study a possible disease overlap. From a methodological point of view, the authors describe the type of study well, specifying that it is a prospective observational study and also specifying the timeline. Furthermore, the authors confirm the correctness of the procedures in terms of data protection, privacy and signing of the informed consent before enrolling the patients in the study. The use of statistical tests is also good, which are in line with the choice of data analysis but certainly with the data in possession and, evaluating the type of study, the authors could have used other statistical techniques to strengthen their initial hypothesis. the presentation of the tables and graphs is good but could be improved. The results show how important the prevalence of NDKD is and how important it is to be able to build an early diagnosis by biopsy in order to adapt the best possible pharmacological treatment and avoid a possible progression of the disease. The advice I give is to perform an analysis of the interaction versus gender variable, as the authors highlight in the descriptive table that 81% of the patients are male and gender could modify or interact with the pathophysiology of the patients studied.

Answer:

  • Thanks a lot for your detailed analysis of the article and your esteemed comments.
  • I agree with the reviewer that 81% of the patient cohort were males. This might be due to the referral bias to the tertiary care centre. However, there was no significant difference between gender distributions in DKD and NDKD patients ( Table-1).